# Comparative Characterization of Volatile Compounds of Ningxiang Pig, Duroc and Their Crosses (Duroc × Ningxiang) by Using SPME-GC-MS

**DOI:** 10.3390/foods12051059

**Published:** 2023-03-02

**Authors:** Bangqiang Zhu, Hu Gao, Fang Yang, Yiyang Li, Qiaoyue Yang, Yinchang Liao, Haimin Guo, Kang Xu, Zhiqiang Tang, Ning Gao, Yuebo Zhang, Jun He

**Affiliations:** 1Hunan Provincial Key Laboratory for Genetic Improvement of Domestic Animal, College of Animal Science and Technology, Hunan Agricultural University, Changsha 410128, China; 2Laboratory of Animal Nutrition Physiology and Metabolism, The Institute of Subtropical Agriculture, The Chinese Academy of Sciences, Changsha 410125, China; 3Ningxiang Animal Husbandry and Fishery Affairs Center, Ningxiang 410600, China

**Keywords:** volatile flavor compounds, tetradecanal, GC-MS, Ningxiang pig, Duroc

## Abstract

With the aim to study the flavor characteristics of Ningxiang pigs (NX), Duroc (DC) pigs, and their crosses (Duroc × Ningxiang, DN), electronic nose and gas chromatography–mass spectrometry analysis were used to detect the volatile flavor substances in NX, DC, and DN (*n* = 34 pigs per population). A total of 120 volatile substances were detected in the three populations, of which 18 substances were common. Aldehydes were the main volatile substances in the three populations. Further analysis revealed that tetradecanal, 2-undecenal, and nonanal were the main aldehyde substances in the three kinds of pork, and the relative content of benzaldehyde in the three populations had significant differences. The flavor substances of DN were similar to that of NX and showed certain heterosis in flavor substances. These results provide a theoretical basis for the study of flavor substances of China local pig breeds and new ideas for pig breeding.

## 1. Introduction

Pork is widely consumed around the world because of its superior characteristics, such as large output, rich nutrition, easy cooking, and widely accepted flavor [1,2]. With the improvement of people’s requirements for pork quality, people pay more and more attention to the flavor of pork. Pork flavor is an important component of meat quality, whose main components are water-soluble and volatile compounds [3]. Recently, with the increasingly mature research methods and technologies of volatile flavor compounds, relevant reports are increasing [4,5]. Gas chromatography–mass spectrometry (GC-MS) is often used for the qualitative and quantitative determination of volatile organic compounds [6]. GC-MS analysis has the advantages of high separation efficiency, high sensitivity, simple quantitative, and strong qualitative ability [7]. In recent years, electronic nose (E-nose) had been proven to be another preferable tool for food quality assessment via olfaction, due to its rapid, easy, reliable, accurate, and non-polluting advantages [8,9]. It is an instrument consisting of a sensor array and a pattern recognition system. Moreover, artificial intelligence technology is adopted in E-nose to simulate human’s sense of smell for comprehensive analysis of the “odor” passing through the instrument, so as to achieve objective evaluation of the sample [10]. Therefore, GC–MS and E-nose can be comparably utilized for odor detection [11] and used to analyze the volatile flavor compounds from the macro and micro aspects [12].

Ningxiang pig (NX), a Chinese native fat-type pig breed, is deeply loved by many consumers because of its high intramuscular fat content, delicious meat, and unique flavor [13]. However, due to the disadvantages such as slow growth, low lean meat percentage, and high feed/meat ratio, the development of related industries has been severely limited. Duroc (DC) is a typical lean-type pig breed, which has superior performance in growth rate, feed conversion ratio, and lean meat percentage [14]. It is often used as the male parent in the hybridization improvement of local pigs in China, but its flavor performance is slightly worse than that of fat-type pig species [15]. The offspring of Duroc and Ningxiang pigs—the Du-Ning Binary Hybrid Pig (DN)—has been greatly improved in growth rate and meat quality.

In order to explore the flavor substances of three different types of pigs and compare the differences of flavor substances among three different kinds of pork, E-nose and GC-MS were used in this study. This study can help to explore the characteristic flavor substances of pig breeds and provide some reference for the research that evaluates crossbreeding effects on the flavor substances.

## 2. Materials and Methods

### 2.1. Sample Collection

Longissimus dorsi muscle samples from DC, NX, and DN (*n* = 34 pigs per population), which had reached at the listing date, were collected. NX and DN samples were collected from the Chu Weixiang Slaughtering and Cutting Plant in Ningxiang City, Hunan Province, and DC samples were obtained from the Tangrenshen Slaughtering and Cutting Plant in Zhuzhou City, Hunan Province. The left carcass weights of NX, DC, and DN were 25–28 kg, 32–49 kg, and 27–33 kg respectively. Pigs were shocked with electricity and then exsanguinated during the slaughter. Afterwards, 20~30 g samples of each pig’s longissimus dorsi between the 7th and 8th ribs were taken within two hours of death, similar to our earlier investigation [16]. Then, the samples were crushed by a grinder (180E-Y, Nail, Cixi, China), packed, and kept at a constant temperature of −80 °C with a steady level of humidity.

### 2.2. E-Nose Analysis

#### 2.2.1. Sample Pretreatment

Before the experiment began, the longissimus dorsi muscle samples were placed in a 4 °C refrigerator for thawing 24 h in advance. Then, we added 1.5 mL saturated NaCl solution (proportion 0.5 mL/g) into 3 g sample and heated it in a water bath at 100 °C for 3 min. The volatile flavor compounds were determined after equilibration for 15 min in a 55 °C water bath (DK-98-II, TAISITE, Tianjin, China).

#### 2.2.2. E-Nose Parameter Setting

E-nose (PEN3, AIRSENSE, Schwerin, Germany) parameters were set as follows: the detection temperature was room temperature and sensor purge gas time was 15 s. The detection and washing times were 60 s and 15 s, respectively. In addition, the flow rate of the sample injection was 100 mL/min.

#### 2.2.3. E-Nose Sensor Characteristics

There are ten sensors in the electronic nose we used. Table 1 shows the sensitivity characteristics of each sensor.

### 2.3. GC-MS Analysis

#### 2.3.1. Sample Pretreatment

After thawing at 4 °C, 5.0 g of the longissimus dorsi muscle sample was weighed, ground, and placed in a closed 15 mL headspace vial. The samples were heated at 100 °C for 30 min, equilibrated at 70 °C for 30 min, and extracted at 100 °C for 50 min. The extraction method was solid-phase microextraction (SPME, Shanghai Ampere Experimental Technology Co., Ltd., Shanghai, China). After extraction, it was put into the injection port of GC-MS (QP2010, SHIMADZU, Kyoto, Japan) and desorbed at 230 °C for 5 min.

#### 2.3.2. GC-MS Conditions

To reduce the noise as much as possible, the employed SPME tip (50 μm, DVB/CAR-PDMS, Supelco, Bellefonte, PA, USA) was aged at 240 °C for 40 min at the injection port of a GC-MS. Chromatographic column: DB-5MS (60 m × 0.25 mm × 0.25 μm, Agilent Technologies (Shanghai) Co., Ltd., Shanghai, China) was used to detect the aroma compounds of the three kinds of pork. GC-MS conditions were according to the procedure described by Gao et al. with modifications [17]. The oven temperature program was as follows: initial temperature 35 °C (held for 6 min), rising to 130 °C at the rate of 4 °C/min (held for 2 min), then heated to 230 °C at the rate of <8 °C/min (held for 5 min). Helium was used as the carrier gas at a flow rate of 2.0 mL/min with the splitless GC inlet mode. The MS fragmentation was performed by electronic impact (EI) mode (ionization energy, 70 eV; source temperature, 230 °C). The transmission line temperature was 250 °C. The acquisition was full-scan mode and mass acquisition range of 30–500 *m*/*z*.

### 2.4. Data Analysis

When a complex sample is separated by GC-MS, it will produce many different peaks in the gas chromatogram and each peak generates a unique mass spectrum used for compound identification. Moreover, the corresponding substances in the chromatograms were determined using the NIST spectral library retrieval system and retained data with substance similarity index greater than 80. Statistical results were analyzed using Microsoft Excel 2016 (Microsoft Corp., Redmond, WA, USA), and outliers were excluded according to the three-sigma principle. The relative contents of each component were calculated by peak area normalization method and classified [18]. Correlation analysis and significance test were firstly carried out on the relative contents of eight substances in each group using R software (version 3.6.1) to analyze the relationship between the categories of substances. The intersection and union sets of all detected substances from all individuals in each population were analyzed. Principal Component Analysis (PCA) was performed using prcomp function, and ggplot2 package was used to visualize the result. Partial least squares discriminant analysis (PLS-DA) was performed using the mixOmics package [19].

## 3. Results

### 3.1. Volatile Flavors of Pork Characterized by E-Nose

The results of E-nose showed (Figure 1) that the sensors R6, R7, R8, and R9 had high response values for the volatile flavor substances in the three populations, indicating that the relative contents of the corresponding methyl, inorganic sulfide, ethanol, aromatic compound, and organic sulfide substances in the three populations were high.

It was found that inorganic sulfide, methyl group, ethanol, aromatic compound, and organic sulfide accounted for relatively large and high contents of total volatile flavor substances in NX pork. The similar results were also found in DN and DC pork. However, the proportion of inorganic sulfide in the volatile substances of DC was the highest among the three populations, while the proportion of ethanol and aromatic compounds in DC was the lowest.

### 3.2. Volatile Flavors of Pork Characterized by GC-MS

#### 3.2.1. The Quantity of Volatile Substances

A total of 120 volatile flavor substances were detected in the three populations by GC-MS. All the detected substances were divided into eight major categories (alcohols, aldehydes, ketones, alkanes, alkenes, esters, furans, and acids) (Appendix A). Eighty-eight volatile substances were detected in NX pork by GC-MS, and 21 substances were shared by all individuals. The quantity and relative content of alcohols, aldehydes, and hydrocarbons were higher than that of other categories (Appendix A). This was consistent with the result of E-nose, which had high response values of R6 and R8 sensors sensitive to methyl, ethanol, and aromatic compounds. In Appendix A, a total of 90 volatile substances were detected in the DC by GC-MS and 33 substances were common to all DC individuals. The amount and relative contents of aldehydes were the most abundant, followed by alcohols. These results were reflected in the results of electronic nose that the response value of R7 sensor sensitive to inorganic sulfide was significantly higher than that of R8 sensor sensitive to methyl. A total of 95 substances were detected in all DN samples, including 40 aldehydes and 18 alcohols (Appendix A). Moreover, aldehydes were the most abundant in the eight substance categories, followed by alcohols and hydrocarbons, which was highly consistent with the response value of R7 sensor sensitive to methyl substances.

#### 3.2.2. Correlation of Eight Major Volatile Flavors

Significantly negative correlations were found between alcohols and aldehydes (r = −0.62, *p* < 0.001), between aldehydes and alkanes (r = −0.58, *p* < 0.001), and between alcohols and acids (r = −0.54, *p* < 0.001) in NX pork. There were also negative correlations between furan and aldehydes (r = −0.47, *p* < 0.01), between furan and acids (r = −0.44, *p* < 0.01), and between alcohols and esters (r = −0.39, *p* < 0.05). However, there were significant positive correlations between alcohols and furans (r = 0.51, *p* < 0.01) and between alkenes and esters (r = 0.47, *p* < 0.01) (Figure 2a).

When compared with the relations of the classified substances in DC pork, we found that there were very significantly negative correlations between alcohols and aldehydes (r = −0.72, *p* < 0.001) and between aldehydes and esters (r = −0.65, *p* < 0.001). Additionally, negative correlations were observed between aldehydes and ketones (r = −0.55, *p* < 0.01), between aldehydes and alkanes (r = −0.54, *p* < 0.01), between ketones and furans (r = −0.50, *p* < 0.01), between alkanes and furans (r = −0.48, *p* < 0.01), between esters and furans (r = −0.53, *p* < 0.01), and between alcohols and furans (r = −0.54, *p* < 0.05). However, we found that there were significantly positive correlations between alcohols and ketones (r = 0.63, *p* < 0.01), between aldehydes and furans (r = 0.70, *p* < 0.01), and between alkanes and esters (r = 0.75, *p* < 0.001) (Figure 2b).

Significantly negative correlations were observed between alcohols and esters (r = −0.72, *p* < 0.001), between alcohols and aldehydes (r = −0.48, *p* < 0.01), between alcohols and acids (r = −0.48, *p* < 0.01), between aldehydes and ketones (r = −0.34, *p* < 0.05), between aldehydes and olefins (r = −0.37, *p* < 0.05), and between aldehydes and furans (r = −0.42, *p* < 0.05). However, there were significantly positive correlations between alcohols and ketones (r = 0.54, *p* < 0.001), between alkanes and esters (r = 0.59, *p* < 0.001), between alcohols and furans (r = 0.34, *p* < 0.05), and between acids and esters (r = 0.38, *p* < 0.05) (Figure 2c).

The relative contents of eight substances in three populations were compared, and we found that aldehyde substances, whether in quantity or relative content, were the most abundant in the three populations and the main volatile flavor substances in the three populations. The multiple comparisons of eight substances among the three populations showed that the aldehydes between NX and DC were significantly different (*p* < 0.05), and the alcohols, alkanes, and acids were significantly different (*p* < 0.01). The differences of alcohols between NX and DN were very significant (*p* < 0.01), while those of other substances were not significant (*p* > 0.05). The differences in alkanes and acids between DN and DC were significant (*p* < 0.01) (Figure 2d).

#### 3.2.3. PCA of Volatile Flavors of Different Pork

The PCA results for total volatiles in NX showed that PC1 and PC2 accounted for 86.7% and 8.4% of the variation, respectively. Moreover, tetradecanal was located at the positive end of PC1 and the negative end of PC2, which was opposite to the position of hexadecal (Figure 3a). PCA analysis of 21 common substances in 34 NX samples showed that both tetradecanal and 2-undecenal were in the negative direction of PC1, and PC1 explained 92.1% of the variation, while PC2 explained 6.5%. These results indicated that tetradecanal, 2-undecenal, trans, trans-2,4-decadienal, and nonanal contributed extremely to the NX volatile flavor (Figure 3b).

The PCA results of 90 volatiles detected in all DC samples showed that trans-2-decenal and tetradecanal were in the opposite position on PC1, with PC1 and PC2 accounting for 73.3% and 6.9% of the variance, respectively (Figure 4a). Moreover, tetradecanal and 2-undecenal were both in the negative position on PC1. In the meantime, PC1 and PC2 accounted for 71.9% and 8.9% of the variance, separately (Figure 4b). Therefore, tetradecanal, 2-undecenal, (E)-2-hepteneal, and nonanal contributed greatly to the volatile flavor in the common substances in DC pork.

The PCA results of 95 volatiles detected in DN showed that PC1 and PC2 accounted for 84.5% and 9% of the variance, respectively. Tetradecanal was the same as hexadecal in the direction of the position on PC1, but in the opposite position on PC2 (Figure 5a). The PCA results of 27 common substances in DN pork showed that PC1 accounted for 93.6% of the variance and PC2 accounted for 4.5%. Tetradecanal was located at the positive end of PC2 and the negative terminal of PC1. Moreover, 2-undecenal was located at the negative terminal of PC2. These results suggested that tetradecanal and 2-undecenal contributed mostly to the DN volatile flavor compounds (Figure 5b).

### 3.3. Comparison of Volatile Flavor Compounds among Three Populations

In order to further compare the differences among the three kinds of pork, the response values of R7, R8, and R9 sensors from three populations were examined with the one-way variance analysis. We found that there were significant differences between NX and DC (*p* < 0.01) and between DN and DC (*p* < 0.05) on the response value of R7 sensor. On the response value of R8 sensor, DC was significantly different from NX (*p* < 0.01) and DN (*p* < 0.05). These results indicated that the proportions of inorganic sulfide and alcohol aromatic compounds in DC were quite different from those of NX and DN.

With the aim to further study the relationship of the flavor substances among the three populations, we analyzed all the detected substances in three populations and found that 62 flavor substances were common to the three populations. In addition to the common substances, the intersections of DC and NX, DN, and DC, and DN and NX contained 1, 11, and 17 substances, respectively. Additionally, NX, DC, and DN contained 8, 16, and 5 population-specific substances, respectively (Figure 6a). Among the 62 common substances, the relative contents of 16 substances in the DN population were higher than those in the NX and DC populations, and 15 substances were lower than those in the other two populations. The one-way variance analysis of these substances showed that the relative contents of (Z)-9-hexadecenal, cyclotetradecane, (E, E)-2,4-nonenal, (E)-2-heptene aldehyde, benzaldehyde, trans-2-octenal, trans-2-decenal, 3-ethyl-2-methyl-1,3-hexadiene in NX and DN were significantly different from those in DC. In addition, the difference of 7-tetradecene-1-ol was extremely significant between DC and NX, and the difference of trans-2-nonanal was significant. The difference between DN and DC in the relative contents of trans-2,4-decadienal and trans-2-nonanal was extremely significant. The difference between DN and NX in 7-etradecene-1-ol and benzaldehyde was extremely significant, and that between (Z)-9-hexadecenal and trans, trans-2,4-decadienal was significant. Benzaldehyde and (Z)-9-hexadecenal showed significant differences among the three populations. (E)-2-hexadecenal was only detected in NX and DC, but there was no significant difference between the two populations. Among the 17 substances only detected in NX and DN, the relative contents of cis-7-tetradecene-1-ol and γ-palmitolactone were significantly different between the two populations. Among the 11 substances specific to DN and DC, the differences of (Z)-7-hexadecenal and 2,4-undecenal between these two populations were very significant, (6Z)6-pentaerythritol 1-ol,9-methylbicyclonane, 5-hexyl-3,3-dimethyl-1-cyclopentene, and octyl formate were significantly different. 6-undecyloxa-2-one, cyclododecene, 1-pentadecene, and (E)-2-(2-penteno) furan in DN were more volatile than that in DC and NX (Appendix A).

The total number of volatile substances in DN was more than two parent population (NX and DC), and the relative contents of alcohols, hydrocarbons, acids, and furans were higher than the average of the corresponding substance contents in NX and DC. In addition, PLS-DA results showed the three populations were easily recognized based on the relative contents of 120 flavor substances. These flavor substances may be the indicators to distinguish these three different kinds of pork. Moreover, DC could be obviously distinguished from NX and DN. The intersection between DN and NX (Figure 6b) indicated that DN, the offspring of NX and DC, were similar to NX in the components and contents of volatile substances.

A total of 18 volatile flavor compounds were shared by all individuals in the three populations, including sixteen aldehydes, one alcohol, and one furan. They were included in the 65 common substances mentioned above. It should be noted that the relative contents of tetradecanal, 2-undecenal, and nonanal in the 18 substances were high, which might make an important contribution to the pork flavor (Figure 6c).

## 4. Discussion

Correlation analysis can be used to study the relationship between volatile substances and determine the statistical relationship between two or more variables [20] to provide a theoretical basis for further analysis. Sulfur-containing amino acids decompose under heat to produce inorganic sulfides and can affect the formation of pyrazines and nitrogen-containing compounds [21]. Electronic nose results showed that DC contained much more sulfur compounds than the other two types of pork. This may indicate to some extent that the content of sulfur-containing amino acids in DC was higher than that in NX and DN. There was no significant difference in electronic nose results between DN and NX, but the proportions of inorganic sulfide and ethanol aromatic compound of DC were significantly different from those of NX and DN (*p* < 0.05), indicating that the volatile substances in NX and DN were not significantly different.

In this study, the aldehydes in NX, DN, and DC populations had the highest content among the eight substances and were the main components of flavor substances, which were consistent with previous research reports [22,23,24]. Aldehyde compounds have low olfactory threshold [25], and can greatly affect the flavor of meat with relatively small content. Therefore, their contribution to flavor substances in meat quality cannot be ignored. Given that NX is a typical fatty pig breed with higher intramuscular fat content than DC and most of the aldehyde substances come from the oxidative [26], this may be the main reason why the relative content of aldehyde substances in NX detected by GC-MS is higher than that of DC.

PCA is one of the commonly used dimensionality reduction techniques [27] to analyze the volatile flavor compounds data after GC-MS quantification and find the main components in multiple variables that affect flavor. Our PCA results showed that tetradecanal, 2-undecenal, and nonanal were the main components of the volatile flavor compounds in the three populations. Tetradecanal has a soft, greasy note with citrus and iris-like notes. It has been reported that tetradecanal is the main flavor compound of the functional sausage [28] and Dongpo pork [29] and also an important volatile compound indicator [30] to distinguish whether the dry cured meat sample has been polluted by ochratoxin. 2-undecenal has aldehyde, wax, citrus, and fat flavors and is naturally present in milk and stewed chicken. Nonanal is an important natural fragrance component used to enhance the fragrance of flowers and the citrus flavor in perfume products [31], and is also an important volatile compound with the unique flavor and fragrance of traditional Chinese chicken soup [26,32]. The key aroma components in the epidermis and breast meat of Beijing roast duck also contain nonanal [33]. The contribution of nonanal to the aroma of thyme stewed mutton is also significant [34]. Therefore, it can be inferred that tetradecanal, 2-undecanal, and nonanal are important factors affecting the flavor of NX, DN, and DC. It has been reported [34,35,36] that hexanal is the main aldehyde substance in pork. However, in our study, the relative contents of hexanal in the DC, DN, and NX populations were 3.38%, 2%, and 1.92%, respectively. It is not the main flavor component of the three pork types in this study. Tetradecanal, 2-undecenal, and nonanal were the main aldehydes in the three populations in this study, which were different from the above reports. Moreover, the correlation between aldehydes and alcohols was negatively (*p* < 0.05) correlated among the three types of pork. It may be caused by the mutual transformation of some substances during heating.

It was reported that benzaldehyde is a key aromatic active compounds of egg-flavored Sachima [37] and related to the formation of characteristic flavor substances in Jinhua ham [38]. Moreover, it can also be used as an indicator of flavor deterioration in yak meat [39]. An appropriate amount of benzaldehyde can improve the flavor of meat quality, but excessive benzaldehyde will have an adverse effect on the flavor of meat quality. There were significant differences in the relative content of benzaldehyde among three kinds of pork in this study. The benzaldehyde content of DC was significantly higher than that of NX. These results suggested that benzaldehyde might be an important factor affecting the three pork flavors.

The relative content of alcohols, hydrocarbons, acids, and furans in DN was higher than the mean of the corresponding substances in NX and DC. Furthermore, the relative content of some volatile substances in DN was higher than that in NX and DC, which include the substances that were not in the other two groups. This showed a certain heterosis. The difference between NX and DN was slight, but it was significant between NX and DC. DN significantly differed from NX only in a fraction of substances. PLS-DA analysis of the volatile flavor compounds revealed that DN exhibited a similar volatile flavor compound to NX. To some extent, it was feasible to improve the flavor performance of offspring by using the complementary advantages of parents. However, there were few reports on the volatile flavor compounds in parents and offspring, so the improvement of meat flavor compounds by hybridization still needs further research.

## 5. Conclusions

In summary, the combination of electronic nose and GC-MS was used to analyze the flavor substances in three populations. It was found that 18 substances were common among all the pigs of the three populations, and the relative content of aldehydes was the most among the 120 volatile substances detected. In addition, PLS-DA results showed the three populations were easily recognized based on the relative contents of 120 flavor substances. Further analysis revealed that tetradecanal, 2-undecenal, and nonanal were the main flavor substances in NX, DN, and DC, and benzaldehyde might be the differential volatile flavor substance in three populations. What is more, the flavor substances of the offspring showed certain heterosis. The findings would provide a reference for the future research on the flavor substances of local and foreign pig breeds, as well as the heterosis of parent and offspring in volatile flavor substances.

## Figures and Tables

**Figure 1 foods-12-01059-f001:**
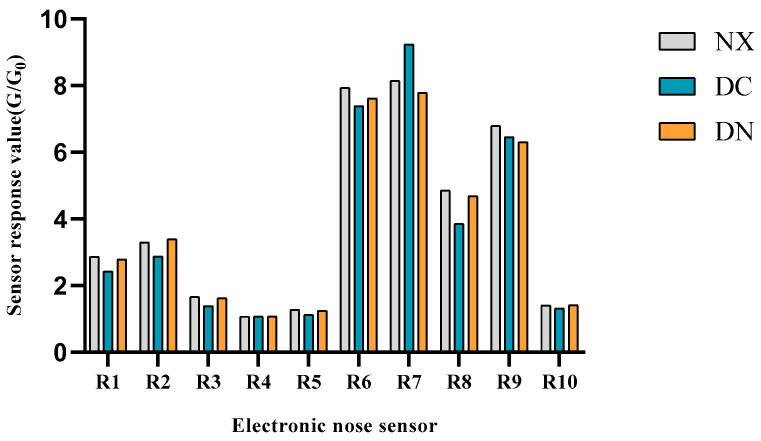
Analysis results of electronic nose in three kinds of pork.

**Figure 2 foods-12-01059-f002:**
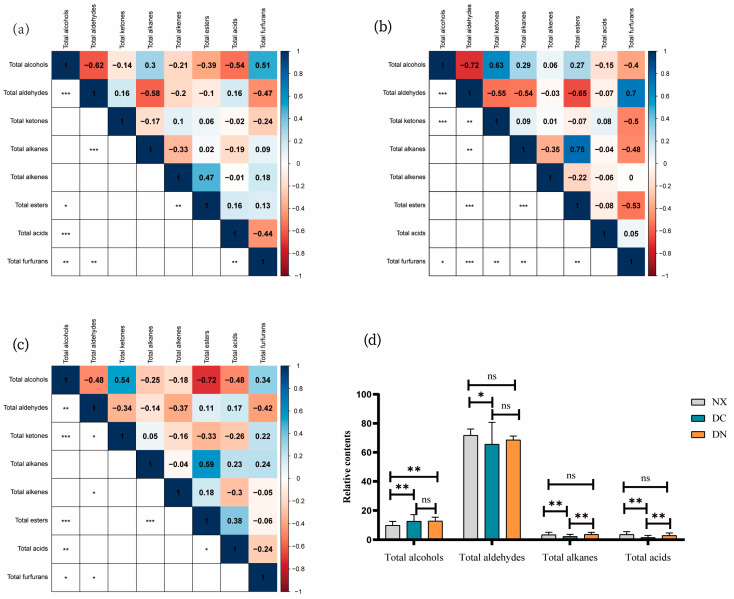
Correlation analysis results for relative content of three kinds of pork. (**a**) Correlation analysis results for relative content of NX eight substances, * (*p* < 0.05), ** (*p* < 0.01), *** (*p* < 0.0001). (**b**) Correlation analysis results for relative content of DC eight substances, * (*p* < 0.05), ** (*p* < 0.01), *** (*p* < 0.0001). (**c**) Correlation analysis results for relative content of DN eight substances, * (*p* < 0.05), ** (*p* < 0.01), *** (*p* < 0.0001). (**d**) Significance analysis of relative levels of alcohols, aldehydes, alkanes, and acids in the three kinds of pork, * (*p* < 0.05), ** (*p* < 0.01), ns (*p* > 0.05).

**Figure 3 foods-12-01059-f003:**
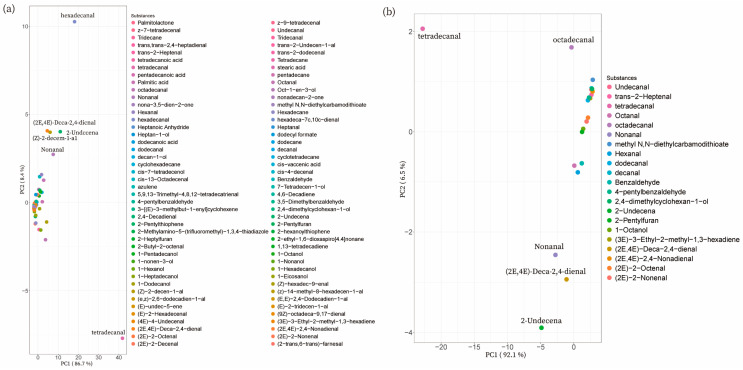
PCA results of the detected substances in NX pork. (**a**) The PCA result of all detected substances in NX pork. (**b**) The PCA result of substances shared by NX samples.

**Figure 4 foods-12-01059-f004:**
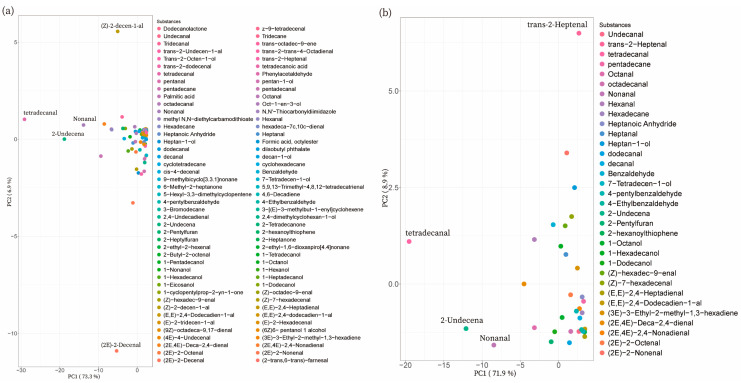
PCA results of the detected substances in DC pork. (**a**) The PCA result of all detected substances in DC pork. (**b**) The PCA result of substances shared by all DC samples.

**Figure 5 foods-12-01059-f005:**
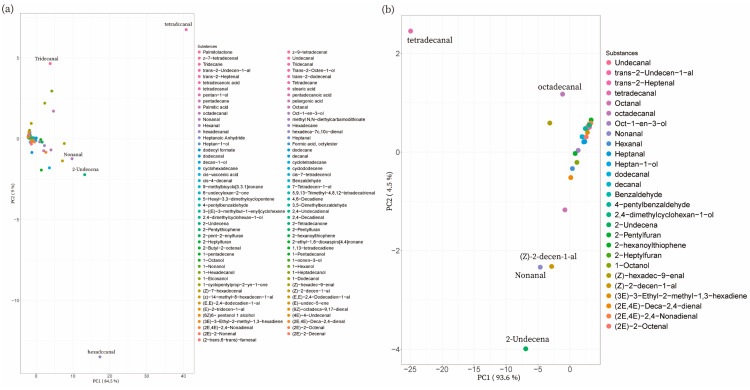
PCA results of the detected substances in DN pork. (**a**) The PCA result of all detected substances in DN pork. (**b**) The PCA result of substances shared by all DN samples.

**Figure 6 foods-12-01059-f006:**
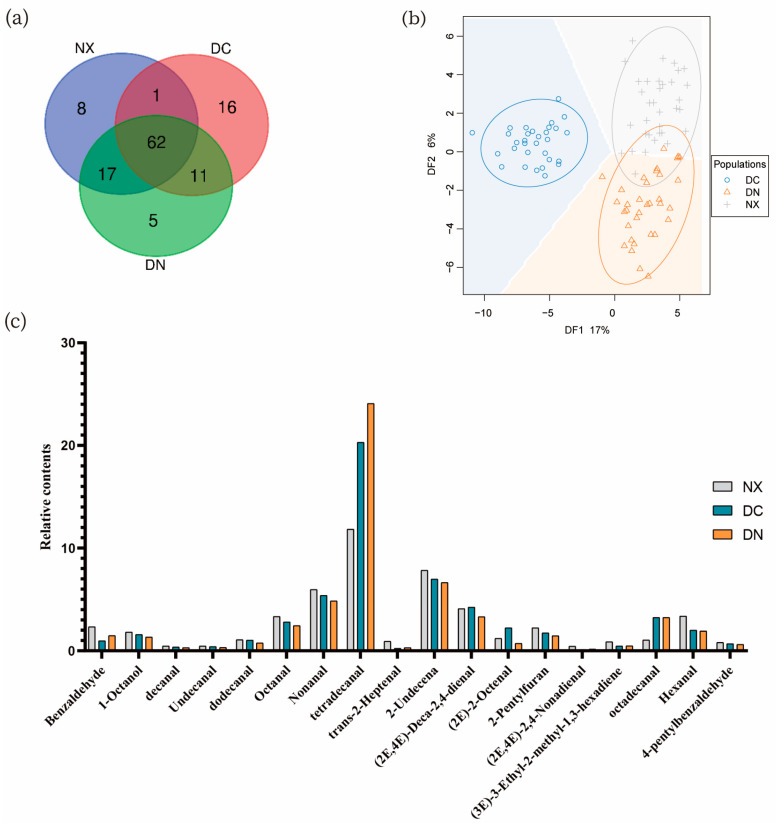
Comparison of the relative contents of flavor substances among three kinds of pork. (**a**) Venn diagram of 120 substances in three kinds of pork. (**b**) Partial least squares discriminant analysis (PLS-DA) of the relative contents of 120 substances in three kinds of pork. (**c**) Relative contents of 18 common volatile flavor substances, which are shared among all samples.

**Table 1 foods-12-01059-t001:** Electronic nose sensor performance description.

NO	Sensor	Sensitive Substance
R1	W1C	Aromatic component
R2	W5S	Oxynitride
R3	W3C	Ammonia, aromatic compound
R4	W6S	Selective to hydrogen
R5	W5C	Alkane and aromatic compound
R6	W1S	Sensitive to methyl
R7	W1W	Inorganic sulfur compound
R8	W2S	Ethanol, aromatic compound
R9	W2W	Organosulfur compound
R10	W3S	Long chain alkane

## Data Availability

The data presented in this study are available from the corresponding authors upon request.

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
