# Peer review of "Comparative Characterization of Volatile Compounds of Ningxiang Pig, Duroc and Their Crosses (Duroc × Ningxiang) by Using SPME-GC-MS"

_foods, 2023, doi:10.3390/foods12051059_

Round 1
Reviewer 1 Report
Referee report for the manuscript foods-2185913
„Comparative Characterization of Volatile Compounds of Ningxiang Pig, Duroc and Their Crosses (Duroc×Ningxiang) by using SPME-GC-MS”
by Bangqiang Zhu, Hu Gao, Fang Yang, Yiyang Li, Qiaoyue Yang, Yinchang Liao, Haimin Guo, Kang Xu, Zhiqiang Tang, Ning Gao, Yuebo Zhang and Jun He
The manuscript is an interesting study on the possibility of characterising and differentiating aroma /volatile compounds from pig meat of the varieties of Ningxiang pig, Duroc pig, and the crossing of the two, Ningxiang×Duroc. Experiments are presented where the response of the Headspace-SPME-GC/MS analysis is compared to the response of an electronic nose.
The authors use a well-established measurement setup for the chromatographic characterisation of food volatiles (HCS-SPME-GC/MS) while the use of an electronic nose adds some novelty to the paper. Chemometric data treatment has been used to identify those compounds which are the most relevant, but since apparently no olfactometric measurements were performed to actually determine the smell / flavour of individual compounds, it is not clear whether the relevance has been judged on the basis of the aroma value of the compounds detected, or merely on the basis of their concentration (which does not have to be correlated to their olfactometric value or impact).
In general, the manuscript is a bit sparse in the description of the experimental procedure(s) and requires significant improvement in this respect. At the current time, important information on how experiments and measurements were performed are missing and need to be added.
For this reason, it is suggested that this manuscript undergoes a major revision.
Some technical comments which also shall be addressed when this manuscript is revised, are listed below.
- L.41: What is “solid-phase microextraction-meteorological chromatography-mas spectrometry” ? Please explain !
- L.62: instead of “quick-freezing” -> shock-frozen
- L.64: the method of sample preparation is extremely important for the development (and assessment) of the flavour. In this case, it appears that the authors cooked the meat (at 100°C for 3 min). This leads to a pattern of flavour compounds that is significantly different from what is observed when the meat is grilled, roasted or fried. This has to be taken into account when comparing these results with results of other authors (who may have applied different conditions for sample preparation).
- L: 70. The manuscript completely lacks the description of the instrumentation used. In particular, nothing is reported on the e-nose that was used for the experiments: Is it a commercially available instrument ? Is it a home-built instrument ? If the former, identify manufacturer and type; if the latter, than provide details about its construction and performance.
- L.74 (e-nose characteristics): The claim of the authors is that “each electronic nose sensor can identify a specific type of substance” - how has this been assessed ? Have the authors performed a selectivity test, or is there a reasonable justification for why the authors believe that one sensor would be specific for “one type of substance” (I assume, I may translate this with one compound class) ? Even if this is so, it appears that some of the sensors do not respond to a single compound class, but to several, such as R8: ethanol, aromatic compound[s]. So, the author’s claim of specificity of the individual sensors is not correct.
- L.79: The authors doe not mention n what type of GC/MS equipment the work was performed. Please add !
- L.84: Also, it is not mentioned what type of SPME device and fibres were used. Identify these unambiguously !
- L. 87: What is the supplier of the chromatographic column ?
- L. 90: The descriprtion of the temperature program is confusing or incorrect: What is meant by: “The initial temperature was maintained at 35°C for 6 min (insert a space !), and gradually increased to 130°C at a rate of 4°C/min (delete the space !) for 2 min (??!?), and finally increased from 130°C to 230°C at a rate of 8 °C/min (delete the space !) for 5 min (??!?). If you consider the first ramp, an increase of 4°C/min for 2 min is only a temperature difference of (°C, but not of 95°C as required between 35 and 130°C. Please clarify or correct ! Furthermore, the description of the injection (= SPME desorption) parameters is missing.
- L.93: It is not (only) the MS fragmentation which is performed by the EI source, but more importantly the ionization.
- L.97: I do not understand the meaning of “The corresponding mass spectra were qualitatively determined for each peak based on computer graphics.” Please clarify !
- L100: The “similarity” is either reported as percentage (“80%”) or as similarity index.
- L.109: What is the supplier (& place of vendor) of the mixOmics package? (or give the reference to the publication describing this R package in PloS One)
- The entire discussion in section “3. Results” raises the question what s the significance of the results: The most abundant compound does not necessarily have to have the biggest contribution or impact on the flavor profile of a sample. The authors mainly discuss their results in terms of (high or low) concentrations, but not of odour impact. Is this relevant then for an olfactometric study and assessment of meat sensory quality?
- Figures 3 and 4 are technical inadequate and must be reworked: The fontsize of the legend is so small that it is practically illegible !
- Can you check with the figure numbers: Does figure 5 appear before figure 4 ?
- Caption of Figure 5: what is meant by “total substances” ?
- L.266: It is not understood on which basis the authors claim the presence of low or high concentrations of particular volatile compounds. I assume that the authors take the response of the e-nose as an indication whether measured concentrations are high or low. However, this assumption must be proven first by a suitable set of reference analyses before it can be assumed that the response of the e-nose is uninterfered and proportional to concentration.
- L.291: The statement “Given that most of the aldehyde substances …” is not clearly understandable. Please reconsider !
- A large part of the discussion under section “4. Discussion” reads more like a literature review, not so much like a discussion. Please adjust both format and contents
- L.349, Conclusion: Does the statement that 18 compounds are common to all three varieties of pig correspond to the same data as presented in figure 5 on p. 7 ? In this figure, 62 substances are mentioned as being common to all three breeds of pig. Please clarify the difference !
- References: for all references the first names have been abbreviated – except for reference [7] – please change this !
Author Response
Response to Reviewer 1 Comments
We would like to thank you for your careful reading, helpful comments, constructive suggestions and professional review work on our article, which has significantly improved the presentation of our manuscript. According to your suggestions, we have made extensive corrections to our previous draf. The detailed corrections are listed below.
Point 1: L.41: What is “solid-phase microextraction-meteorological chromatography-mas- spectrometry”? Please explain!
Response 1: We seriously apologize for this carelessness and thank you so much for pointing out this issue. The full name of HS-SPME-GC-MS is headspace solid-phase microextraction gas chromatography/mass spectrometry.
Point 2: L.62: instead of “quick-freezing” -> shock-frozen
Response 2: We sincerely thank you for careful reading. The manuscript has been checked thoroughly again and some grammar errors were corrected.
Point 3: L.64: the method of sample preparation is extremely important for the development (and assessment) of the flavour. In this case, it appears that the authors cooked the meat (at 100°C for 3 min). This leads to a pattern of flavour compounds that is significantly different from what is observed when the meat is grilled, roasted or fried. This has to be taken into account when comparing these results with results of other authors (who may have applied different conditions for sample preparation).
Response 3: We completely agree with you. We heated the meat in a 100℃ water bath for 3 minutes, mainly to study the general categories of flavour substances in cooked meat without adding other additives. This approach does differ from some other studies and we took it into account when comparing the results. The citations in this section were used to illustrate the applicability of GC-MS to the determination of flavour substances. Thank you again for pointing this out.
Point 4: L: 70. The manuscript completely lacks the description of the instrumentation used. In particular, nothing is reported on the e-nose that was used for the experiments: Is it a commercially available instrument ? Is it a home-built instrument ? If the former, identify manufacturer and type; if the latter, then provide details about its construction and performance.
Response 4: We apologize for the lack of information. The e-nose is a commercially available instrument and we have supplemented its information at line 85.
Line 85
“E-nose (PEN3, AIRSENSE, Germany) parameters were set as follows: The detection temperature was 25℃ and sensor purge gas time was 15 s. The detection and washing times were 60 s and 15 s, respectively. In addition, the flow rate of sample injection was 100 ml/min”
Point 5: L.74 (e-nose characteristics): The claim of the authors is that “each electronic nose sensor can identify a specific type of substance” how has this been assessed ? Have the authors performed a selectivity test, or is there a reasonable justification for why the authors believe that one sensor would be specific for “one type of substance” (I assume, I may translate this with one compound class) ? Even if this is so, it appears that some of the sensors do not respond to a single compound class, but to several, such as R8: ethanol, aromatic compound[s]. So, the author’s claim of specificity of the individual sensors is not correct.
Response 5: Our previous description of the characteristics of the sensor is indeed incorrect. The sensors we use don't recognize specific substances. We removed the previous incorrect description at line 90.
Point 6: L.79: The authors do not mention what type of GC/MS equipment the work was performed. Please add!
Response 6: Thanks, we added its information as follows:
Line 100-102
“After extraction, it was put into the injection port of GC–MS (QP2010, SHIMADZU, Japan) and desorbed at 230 °C for 5 min.”
Point 7: L.84: Also, it is not mentioned what type of SPME device and fibres were used. Identify these unambiguously!
Response 7: We supplemented relevant information at line 104-106.
Line104-106
“To reduce the noise as much as possible, the employed SPME tip (50 μm, DVB/CAR-PDMS, Supelco, Bellefonte, PA, USA) was aged at 240°C for 40 min at the injection port of a GC-MS.”
Point 8: L. 87: What is the supplier of the chromatographic column?
Response 8: Thanks, we added the missing information about the supplier of the chromatographic column at line 106-108.
Line 106-108
“Chromatographic column: DB-5MS (60 m × 0.25 mm × 0.25 μm, Agilent Technologies (Shanghai) Co., Ltd.) was used to detect the aroma compounds of the three kinds of pork.”
Point 9: L. 90: The description of the temperature program is confusing or incorrect: What is meant by: “The initial temperature was maintained at 35°C for 6 min (insert a space !), and gradually increased to 130°C at a rate of 4°C/min (delete the space !) for 2 min (??!?), and finally increased from 130°C to 230°C at a rate of 8 °C/min (delete the space !) for 5 min (??!?). If you consider the first ramp, an increase of 4°C/min for 2 min is only a temperature difference of (°C, but not of 95°C as required between 35 and 130°C. Please clarify or correct ! Furthermore, the description of the injection (= SPME desorption) parameters is missing.
Response 9: Thanks. We rephrased these sentences and supplemented the description of the injection. Please find below the updated sentences in the revised manuscript.
Line 108-112
“GC–MS conditions were according to the procedure described by Gao et al. with modifications. The oven temperature program was as follows: initial temperature 35 °C (held for 6 min), rising to 130 °C at the rate of 4 °C/min (held for 2 min), then heated to 230 °C at the rate of 8 °C/min (held for 5 min). Helium was used as the carrier gas at a flow rate of 2.0 mL/min with the splitless GC inlet mode.”
Line 100-102
“After extraction, it was put into the injection port of GC–MS (QP2010, SHIMADZU, Japan) and desorbed at 230 °C for 5 min.”
Point 10: L.93: It is not (only) the MS fragmentation which is performed by the EI source, but more importantly the ionization.
Response 10: We fully agree with you and thanks for your valuable comments. We made a modification at line 112-115.
Line 112-115
“The MS fragmentation was performed by electronic impact (EI) mode (ionization energy, 70 eV; source temperature, 230 °C). The transmission line temperature was 250 °C. The acquisition was full-scan mode and mass acquisition range of 30–500 m/z.”
Point 11: L.97: I do not understand the meaning of “The corresponding mass spectra were qualitatively determined for each peak based on computer graphics.” Please clarify!
Response 11: We rephrased this sentence as “When a complex sample is separated by GC-MS, it will produce many different peaks in the gas chromatogram and each peak generates a unique mass spectrum used for compound identification.” (Line 124-126)
Point 12: L100: The “similarity” is either reported as percentage (“80%”) or as similarity index.
Response 12: Thanks, we have changed it to similarity index.
Point 13: L.109: What is the supplier (& place of vendor) of the mixOmics package? (or give the reference to the publication describing this R package in PloS One)
Response 13: Thanks. The following reference describing this R package was added.
“Rohart, F., Gautier, B., Singh, A., & Lê Cao, K. A. (2017). mixOmics: An R package for 'omics feature selection and multiple data integration. PLoS computational biology, 13(11), e1005752. https://doi.org/10.1371/journal.pcbi.1005752”
Point 14: The entire discussion in section “3. Results” raises the question what’s the significance of the results: The most abundant compound does not necessarily have to have the biggest contribution or impact on the flavor profile of a sample. The authors mainly discuss their results in terms of (high or low) concentrations, but not of odour impact. Is this relevant then for an olfactometric study and assessment of meat sensory quality?
Response 14: We fully agree with your opinion that the most abundant compounds do not necessarily have the greatest contribution or effect on the flavor profile of the sample. In this study, we were aimed to studied the similarities and differences of flavor substance content among the three different kinds of pork. There are few studies on the flavor compounds in Ningxiang pigs. Through this study, we can find out the main flavor substances in three kinds of pork, although they may not contribute much to the flavor, but it can provide reference for the research in this area and help for the collection and analysis of the aromatic value of the later flavor substances. Thank you again for your constructive comments.
Point 15: Figures 3 and 4 are technical inadequate and must be reworked: The fontsize of the legend is so small that it is practically illegible!
Response 15: Thank you very much for your comments. We have made modifications to the problems.
Point 16: Can you check with the figure numbers: Does figure 5 appear before figure 4?
Response 16: For our negligence, we are deeply sorry and grateful for your care and responsibility. We have modified the order of the charts.
Point 17: Caption of Figure 5: what is meant by “total substances”?
Response 17: Thanks. To make it clearer, we changed it to “Venn diagram results of 120 substances in three kinds of pork”.(Line 244)
Point 18: L.266: It is not understood on which basis the authors claim the presence of low or high concentrations of particular volatile compounds. I assume that the authors take the response of the e-nose as an indication whether measured concentrations are high or low. However, this assumption must be proven first by a suitable set of reference analyses before it can be assumed that the response of the e-nose is uninterfered and proportional to concentration.
Response 18: Sorry for confusing you. The relative concentrations of flavor substances were determined by GC-MS. Based on these data, we compared the similarities and differences of the three kinds of pork.
Point 19: L.291: The statement “Given that most of the aldehyde substances …” is not clearly understandable. Please reconsider!
Response 19: Sorry for your confusion. We rephrased this sentence as “Given that NX is a typical fatty pig breed with higher intramuscular fat content than DC and most of the aldehyde substances come from the oxidative, this may be the main reason why the relative content of aldehyde substances in NX detected by GC-MS is higher than that of DC.”
Thanks, we have modified as follows in line 330: “Given that NX is a typical fatty pig breed with higher intramuscular fat content than DC and most of the aldehyde substances come from the oxidative [21], this may be the main reason why the relative content of aldehyde substances in NX detected by GC-MS is higher than that of DC.”
Point 20: A large part of the discussion under section “4. Discussion” reads more like a literature review, not so much like a discussion. Please adjust both format and contents.
Response 20: Thanks for your suggestion. We have adjusted the content and format of the discussion section.
Point 21: L.349, Conclusion: Does the statement that 18 compounds are common to all three varieties of pig correspond to the same data as presented in figure 5 on p. 7 ? In this figure, 62 substances are mentioned as being common to all three breeds of pig. Please clarify the difference!
Response 21: Sorry for confusing you. We do not have a clear enough description in this regard. A total of 62 substances were detected in the three pig populations, and only 18 substances are shared by all individuals of the three population. Therefore, the statement that 18 compounds are common to all three varieties of pig correspond to the same data as presented in figure 4d. For the convenience of readers, we rephrased this sentence as “It was found that 18 substances are common among all the pigs of the three populations”
Point 22: References: for all references the first names have been abbreviated – except for reference [7] – please change this!
Response 22: Thanks. We have modified.
We tried our best to improve the manuscript and made some changes marked in red in our revised manuscript which not influence the content and framework of the paper. We appreciate for Editors/Reviewers’ warm work earnestly, and hope the correction will meet with approval. Once again, thank you very much for your comments and suggestions.

Reviewer 2 Report
a Few suggestions:
- Could you please explain in deep the novelty of the use of this technology and also if will be possible to apply it not limited to chinese market and breeding
- line 59: how were these samples selected? why were these chosen? please provide more details on this.
- line 66: have all the samples been in the fridge for 24 hours? so they were all made on the same day? What if the samples are made months later?
- line 70: Did you expect the sample to stabilize at 25 degrees? from 4 to 25 degrees for each sample how long? how long does the measure last? where were the samples kept?
Information is missing, perhaps a schematic would be recommended
line 74
what sensors are they? did you make them? Are they commercial sensors? how were they chosen? the working temperatures? what data can I produce?
this part must absolutely be increased
information on which GC was used and the type of electronic nose is completely missing? did you do it in the lab? What's your name ? what parts is it made up of?
the article is not bad but some very important information is missing to allow you to understand the results obtained. We kindly ask you to complete the missing parts, providing the missing information
Author Response
Response to Reviewer 2 Comments
Thank you for your nice comments on our article. According to your suggestions, we have supplemented several data here and corrected several mistakes in our previous draft. Based on your comments, we also attached a point-by-point letter to you. We have made extensive revisions to our previous draft. The detailed point-by-point responses are listed below.
Point 1: Could you please explain in deep the novelty of the use of this technology and also if will be possible to apply it not limited to Chinese market and breeding.
Response 1: Thanks for your suggestion. GC–MS is an advanced technology and widely used to qualitative and quantitative analysis for volatile compositions. E-nose designed as an apparatus to mimic the human olfactory perception, can be an innovative measurement system. In recent years, E-nose with rapid, easy, reliable, accurate, and non-polluting advantages has been proven to be another preferable tool for food quality assessment via olfaction. Recently, owing to the advantages of rapid, accurate and effective determination, well verification and complement to each other, E-nose combined with GC–MS has already been used in the analysis of plants, such as bayberry, jujubes and Goji berries. However, little information has been reported in flavor substance analysis of different pig varieties by using E-nose combined with GC–MS. In this study, E-nose coupled with GC–MS were used to detect volatile compositions of Ningxiang pig (a Chinese native fat-type pig breed), Duroc pig (a typical lean-type pig breed), and their hybrid offspring. Therefore, we used this combined technology not only to analyze different pig varieties, but also to innovatively access crossbreeding effect. The technology has no geographical limitations and can be used in global market and breeding.
Point 2: line 59: how were these samples selected? why were these chosen? please provide more details on this.
Response 2: Thanks for your helpful comments. These samples were randomly selected from slaughterhouses. The left carcass weights of NX, DC and DN were 25-28 kg, 32-49kg and 27-33 kg respectively. The reason why we chose these samples is that NX is a typical fat type pig with local characteristics in China, and DC is a typical lean type pig. Through studying the pork flavor substances of them and their hybrid offspring, we can help to develop the economic value of NX and understand the differences of flavor substances between fat-type and lean-type pigs. Thanks for your suggestions, we changed the previous sample collection details as follows:
“NX samples were collected from the Chu Weixiang Slaughtering and Cutting Plant in Ningxiang City, Hunan Province, DC and DN pork samples were obtained from the Tangrenshen Slaughtering and Cutting Plant in Zhuzhou City, Hunan Province. The left carcass weights of NX, DC and DN were 25-28 kg, 32-49 kg and 27-33 kg respectively. Pigs were shocked with electricity and then exsanguinated during the slaughter. Afterwards, 20~30 g samples of each pig’s longissimus dorsi between the sixth and eleventh ribs were taken within two hours of death, similar to our earlier investigation [11]. Then the samples were crushed by a grinder (180E-Y, Nail, Cixi, China), packed and kept at a constant temperature of −80°C with a steady level of humidity.”
Point 3: line 66: have all the samples been in the fridge for 24 hours? so they were all made on the same day? What if the samples are made months later?
Response 3: Thanks for your constructive suggestion. Regarding the storage and preparation time of samples, all the samples were placed in a 4℃ refrigerator for thawing 24 hours in advance. They were all made on the same day. We have not done any experiments about the samples are made months later, and the effect is not very clear.
Point 4: line 70: Did you expect the sample to stabilize at 25 degrees? from 4 to 25 degrees for each sample how long? how long does the measure last? where were the samples kept? Information is missing, perhaps a schematic would be recommended
Response 4: Thank you for pointing this out. 25℃ (room temperature) is the temperature that the sample measured by the electronic nose, not the sample rising from 4 degrees to 25 degrees. The pretreatment of samples before determination by electronic nose is as follows: The longissimus dorsi muscle samples were placed in a 4℃ refrigerator for thawing 24 hours in advance. Then we added 1.5 ml saturated NaCl solution (proportion 0.5 mL/g) into 3 g sample and heated it in a water bath at 100 ℃ for 3 min. The volatile flavor compounds were determined after equilibration for 15min in a 55 ℃ water bath (DK-98-Ⅱ, TAISITE, Tianjin, China). To more readable, we changed the 25℃ to room temperature.
The specific process is as follows:
Point 5: line 74: what sensors are they? did you make them? Are they commercial sensors? how were they chosen? the working temperatures? what data can I produce? information on which GC was used and the type of electronic nose is completely missing? did you do it in the lab? What's your name ? what parts is it made up of ?
Response 5: Missing information has been added in line 85-88.
Line 85-88
“E-nose (PEN3, AIRSENSE, Germany ) parameters were set as follows: The detection temperature was room temperature and sensor purge gas time was 15 s. The detection and washing times were 60 s and 15 s, respectively. In addition, the flow rate of sample injection was 100 ml/min.”
We have added the issue of missing GC-MS information in line 104.
Line 104-118
” To reduce the noise as much as possible, the employed solid-phase microextraction tip (50 μm, DVB/CAR-PDMS, Supelco, Bellefonte, PA, USA ) was aged at 240°C for 40 min at the injection port of a GC-MS (QP2010, SHIMADZU, Japan). Chromatographic column: DB-5MS (60 m × 0.25 mm × 0.25 μm, Agilent Technologies (Shanghai) Co., Ltd.) was used to detect the aroma compounds of the three kinds of pork. GC–MS conditions were accord-ing to the procedure described by Gao et at with modifications [12]. The oven temperature programme was as follows: initial temperature 35°C (held for 6 min), rising to 13°C at the rate of 4°C/min (held for 2 min), then heated to 230°C at the rate of 8°C/min (held for 5 min). Helium was used as the carrier gas at a flow rate of 2.0 mL/min with the splitless GC inlet mode. The MS fragmentation was performed by electronic impact (EI) mode (ion-ization energy, 70 eV; source temperature, 230°C). The transmission line temperature was 250°C. The acquisition was full-scan mode and mass acquisition range of 30–500 m/z”
Electronic nose is a simulated biological olfactory system, which consists of sensor array combined with pattern recognition system.
We tried our best to improve the manuscript and made some changes marked in red in revised paper which will not influence the content and framework of the paper. We appreciate for Editors/Reviewers’ warm work earnestly, and hope the correction will meet with approval. Once again, thank you very much for your comments and suggestions.
